# No Evidence of Disease: Clinically-Risky Adversarial Chest CT Report Generation

**Samra Irshad**[1]  ⬤                                    SAMRA@KHU.AC.KR
[1] *School of Computing, Kyung Hee University, Gyeonggi-do 17104, South Korea*

**Junho Kim**[2]                                              ARKIMJH@ILLINOIS.EDU
[2] *University of Illinois Urbana–Champaign, Champaign, IL, United States*

**Seong Tae Kim**[1,*]                                        ST.KIM@KHU.AC.KR
[1] *School of Computing, Kyung Hee University, Gyeonggi-do 17104, South Korea*

**Editors:** Accepted for publication at MIDL 2026

## Abstract

Automated chest CT radiology report generation has equipped clinicians with the ability to automatically describe clinical findings and abnormalities from CT scans. Given that patient prognosis relies heavily on these reports, generating an accurate CT report is critical. Advances in Multimodal Large Language Models (MLLMs) have enabled substantial improvements in CT-to-text report generation models, yet recent studies show that MLLMs are highly susceptible to adversarial perturbations. Beyond this known susceptibility, it remains unclear what triggers clinically dangerous attack scenarios during medical report generation. Understanding such threats is essential for developing robust medical AI systems — without a clear characterization of the threat, it is challenging to mitigate real-world risks. In this paper, we investigate how chest CT report generation models can be adversarially manipulated and what constitutes an adversarial CT report. We introduce **Clinically Risky Adversarial Report Generation (CRA-RG)**, a threat model that defines clinically realistic adversarial alterations to chest CT reports. To instantiate this threat model, we develop a targeted multimodal attack that perturbs both CT volumes and conditioning text prompts to induce clinically risky changes in reports. We show that our attack can successfully omit and fabricate clinically grounded high-risk CT chest findings (e.g., *nodules* or *lesions*). To the best of our knowledge, our study is the first empirical demonstration that state-of-the-art CT report generation models can be deceived into producing harmful clinical decisions, potentially leading to missed diagnoses or unnecessary biopsies. We evaluate our attack on two state-of-the-art CT report generation models using the publicly available chest 3D CT RadGenome dataset.

**Keywords:** Multimodal Large Language Models, CT Report Generation, Adversarial Attack.

## 1. Introduction

Examining chest scans for abnormalities across thoracic organs and then articulating the observed findings in a detailed written report are core responsibilities of radiologists. Chest radiology reports typically provide structured, organ-based descriptions of both normal anatomy and pathological findings. Given that the interpretation of chest radiographs required to write radiology reports is time-consuming and depends on specialized expertise,

---

* Corresponding author.

there has been growing interest in developing automated systems to assist with chest radiology reporting (Everlight, 2025; Chen et al., 2020; Li et al., 2023b; Zhang et al., 2025a). Automated models for radiology report generation are built on recent advances in multimodal learning, driven mainly by Multimodal Large Language Models (MLLMs) (Li et al., 2023a; Hu et al., 2024; Jian et al., 2024). While prior studies have predominantly focused on chest X-ray report generation, chest CT scans contain much finer anatomical detail and impose a higher interpretive burden on radiologists. As a result, the development of radiology report generation models for chest CT interpretation has only recently begun to receive attention (Chen et al., 2025; Hamamci et al., 2024).

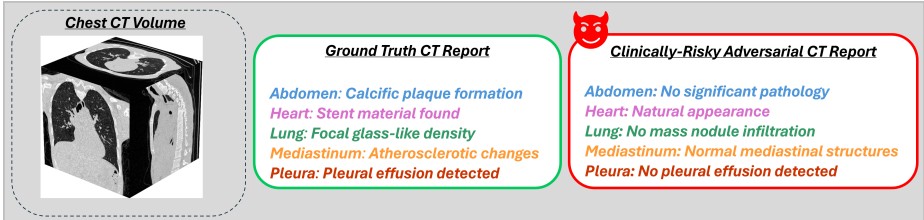

Figure 1: Adversarially manipulated chest CT report: CT volume (left) is shown with the groundtruth CT report (middle) illustrating organ-wise findings, and the corresponding adversarial report (right). Only key phrases are displayed for brevity.

Despite these successes, it is well established that vision models are vulnerable to adversarial attacks (Goodfellow et al., 2015). Although imperceptible, these adversarial attacks can profoundly distort model outputs, making them particularly dangerous in safety-critical domains (Nguyen et al., 2025). Beyond vision models, recent studies have raised new AI safety concerns by demonstrating that even LLMs are vulnerable, with adversarial prompts capable of bypassing built-in guardrails and manipulating models to generate harmful or factually incorrect responses (Han et al., 2024; Zou et al., 2023; Andriushchenko et al., 2025). Because MLLMs rely on LLMs as their language decoder and additionally incorporate visual inputs, the safety weaknesses of LLMs naturally extend to the multimodal setting. As a result, adversarial prompts introduced through either text or images present a new and significant threat to MLLMs (Aafaq et al., 2021; Zhao et al., 2023; Dong et al., 2023; Shayegani et al., 2024). This poses a significant risk to medical AI, particularly multimodal radiology report generation systems that rely on both image and language inputs and may produce clinically dangerous errors when exposed to adversarial prompts or perturbed images. Understanding these attack vectors is therefore essential for developing robust medical AI systems. Motivated by these concerns, we pose the following question:

*Can chest CT radiology report generation models be adversarially attacked, and, more importantly, what does it mean for a radiology report itself to be considered 'attacked'?*

In this paper, we introduce **Clinically Risky Adversarial Report Generation (CRA-RG)**, a new adversarial threat that defines how MLLM-based chest CT-to-text report generation models can be attacked at test time to conceal or fabricate critical chest

findings. We present an example of an Adversarial CT report in Figure 1, illustrating the manipulation of critical chest features. To realize this threat model, we develop a multimodal targeted adversarial attack that leverages learnable visual and textual perturbations and injects them into the model's inputs. Specifically, we apply voxel-level adversarial perturbations to both the full chest CT volume and the anatomy of interest (*e.g.,* lung parenchyma or breast region), as well as perturbations to the text prompt. By jointly attacking the visual and textual representations, our method generates adversarial chest CT reports that remain clinically plausible yet omit and insert high-risk chest CT findings. Our proposed attacking framework is shown in Figure 2.

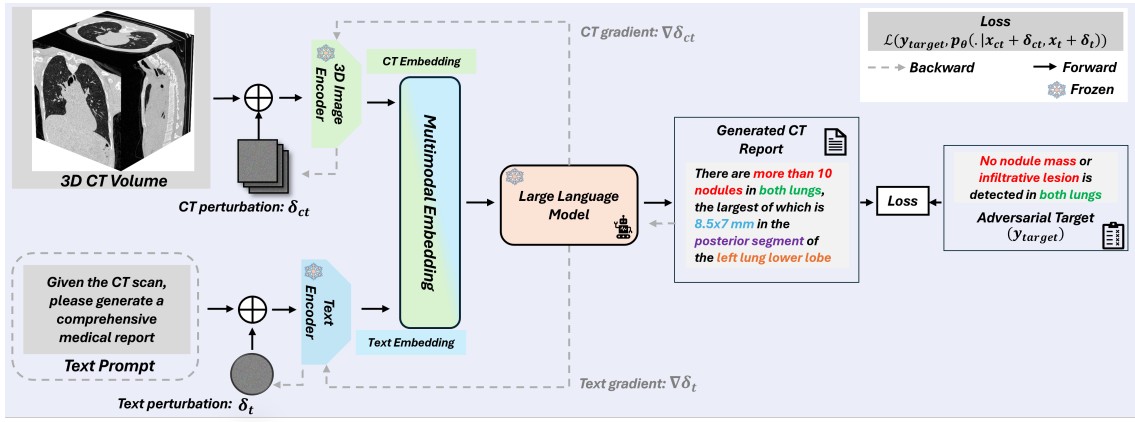

Figure 2: Our proposed framework for Clinically-Risky adversarial CT report generation. Given a 3D chest CT volume and text input at test time, we inject learnable targeted multimodal perturbations $\delta_{ct}$ and $\delta_t$ into CT volume and text prompt embeddings, respectively, to induce clinically risky report changes.

## 2. Related Work

### 2.1. Multimodal Large Language Models (MLLMs)

Typically, MLLMs integrate a visual encoder with a pretrained Large Language Model (LLM) decoder via a connector to learn unified vision–language representations (Liu et al., 2023; Li et al., 2023a; Alayrac et al., 2022). The visual encoder extracts image embeddings, the connector transforms these embeddings into the language module's latent space, and finally, the language decoder generates the textual response conditioned on multimodal prompts. These representations, in turn, provide the foundation for effective cross-modal reasoning (Touvron et al., 2023; Jiang et al., 2023; Achiam et al., 2023). With access to large-scale text corpora organized in natural-language instruction formats and equipped with large parameter-sized models, LLMs acquire strong instruction-following capabilities. When serving as the language decoder in MLLMs, these capabilities enable the model to reason over visual inputs and generate textual responses when guided by multimodal prompts. Leveraging these advances, recent studies have applied MLLMs to radiology

report generation by fine-tuning them on paired radiology scan–report datasets (Shruthi et al., 2024; Lee et al., 2025).

## 2.2. Adversarial Attacks on Multimodal Large Language Models (MLLMs)

Recently, several works have reported that MLLMs are vulnerable to adversarial attacks (Qi et al., 2024; Carlini et al., 2023; Gong et al., 2025). These attacks generally fall into two main categories, **jailbreak** and **evasion** attacks (Qi et al., 2024; Cui et al., 2023). **Jailbreak attacks** are primarily designed to break the safety alignment of MLLMs. On the other hand, **evasion attacks** aim to manipulate visual, textual, or both inputs to alter the model's normal behaviour. Evasion attacks are typically designed to introduce targeted mispredictions or produce untargeted misleading outputs while keeping perturbations small (Cui et al., 2023; Zhao et al., 2023; Hanif et al., 2025). Our work falls into the category of targeted evasion attacks. A few studies have recently examined the adversarial robustness of MLLMs for medical VQA tasks. (Clusmann et al., 2025; Hanif et al., 2025). However, the existing studies attack the MLLMs by assuming a generic target, *e.g., no signs of disease* (Hanif et al., 2025). Such generic targets are insufficient for radiology report generation, which follows a strict organ-based structure. Radiology chest CT reports consist of organ-specific sections, each describing anatomy-specific findings (Hamamci et al., 2024). Each organ-specific section describes the associated abnormalities that are organ-specific. For example, lung findings and abdominal findings involve distinct pathologies and therefore, rely on terminology unique to their respective anatomies. Consequently, a single generic target is insufficient for attacking radiology report generation models. To our knowledge, no prior work has investigated targeted, organ-aware adversarial attacks on CT report generation models. Our method addresses this gap by explicitly modeling and attacking organ-specific findings in CT reporting MLLMs.

## 3. Proposed Methodology

### 3.1. Problem Formulation

Consider a 3D chest CT scan $x_{\mathrm{ct}}$ and a text prompt $x_{\mathrm{t}}$ that instructs the MLLM to generate a text report. At test time, given $(x_{\mathrm{ct}}, x_{\mathrm{t}})$, the model autoregressively predicts the conditional probability of a report $y = (y_1, \ldots, y_K)$, as expressed in Equation (1).

$$p_\theta(y \mid x_{ct}, x_t) = \prod_{k=1}^{K} p_\theta(y_k \mid x_{ct}, x_t, y_{<k}),\qquad(1)$$

where $p_\theta$ denotes the CT-grounded radiology report generation model parameterized by $\theta$, $y_k$ is the $k$-th token in the report, $y_{<k} = (y_1, \ldots, y_{k-1})$ denotes all previously generated tokens, and $K$ is the length of the report. Then, the task of adversarial chest CT report generation $y_{\mathrm{adv}}$ can be formulated as below:

$$p_\theta(y_{\mathrm{adv}} \mid x_{\mathrm{ct}} + \delta_{\mathrm{ct}},\, x_{\mathrm{t}} + \delta_{\mathrm{t}}) = \prod_{k=1}^{K} p_\theta(y_{\mathrm{adv},k} \mid x_{\mathrm{ct}} + \delta_{\mathrm{ct}},\, x_{\mathrm{t}} + \delta_{\mathrm{t}},\, y_{\mathrm{adv},<k}).\qquad(2)$$

In Equation (2), $\delta_{\text{ct}}$ represent the learnable visual perturbation applied to the 3D chest CT volume and is bounded by perturbation magnitude $||\delta_{\text{ct}}||$. The goal of the adversary is to keep the perturbation magnitude minimum ($||\delta_{\text{ct}}|| \leq \epsilon_{\text{ct}}$) to impose imperceptibility between clean image $x_{\text{ct}}$ and perturbed image $x_{\text{ct}} + \delta_{\text{ct}}$. Likewise, $\delta_{\text{t}}$ is the prompt perturbation bounded by $\epsilon_{\text{t}}$ ($||\delta_{\text{t}}|| \leq \epsilon_{\text{t}}$).

To examine whether effective adversarial perturbations can be learned for multimodal inputs that mislead the MLLM into altering high-risk chest findings and subsequently generating a clinically dangerous report, we define a target text that the attacker aims to force the model to generate. This type of attack falls under the category of **targeted adversarial attack**, in which the adversary specifies in advance the exact output (typically known as **target**) they aim to obtain from the victim model. In targeted adversarial attack scenarios, the target is typically drawn from the model's normal output distribution but is deliberately selected because it represents an incorrect or dangerous outcome (Sato et al., 2020). Following this principle, we define clinically plausible target outputs that intentionally contradict the ground truth. These opposing targets — such as reporting an abnormality as absent or fabricating a nonexistent one — are chosen to induce harmful clinical decisions.

### 3.2. Threat Model

We assume a white-box threat model in which the adversary has complete knowledge about the victim CT-to-text generative model $p_\theta$. Under this setting, we focus on learning perturbations for targeted adversarial manipulation of the generated text report.

### 3.2.1. CLINICALLY-RISKY ADVERSARIAL REPORT GENERATION

In our proposed attack, the adversary aims to deceive the MLLM into producing a predefined harmful target report. We sample the targets from the original reports to ensure that $y_{\text{target}} \in \mathcal{Y}$, where $\mathcal{Y}$ denotes the report sentences extracted from the dataset. Thus, the goal of our targeted attack is to drive the MLLM toward generating the predefined target report by minimizing the text modeling loss $\mathcal{L}$:

$$\min_{\delta_{\text{ct}}, \delta_t} \mathcal{L}(y_{\text{target}}, p_\theta(y_{adv} \mid x_{\text{ct}} + \delta_{\text{ct}}, x_t + \delta_t)) \quad \text{s.t.} \quad ||\delta_{\text{ct}}|| \leq \epsilon_{\text{ct}}, ||\delta_t|| \leq \epsilon_t. \tag{3}$$

Each chest CT report describes the thoracic organs and indicates whether abnormalities are present. From these descriptions, we derive candidate negative $\mathcal{Y}_{\text{neg}}$ and positive targets $\mathcal{Y}_{\text{pos}}$. For instance, a negative lung target might be: *No nodule or infiltrative lesion is observed in the lung parenchyma.* Driven by the possible ways in which a chest CT report can be adversarially manipulated, we define the following adversarial goals:

**(a) Suppressing high-risk chest abnormalities:** Here, the adversary aims to increase the likelihood of clinical misdiagnosis by omitting specific targeted abnormalities. We sample the target text from the set of negative sentences in the ground-truth reports, which indicate the absence of any critical abnormality ($y_{\text{target}} \in \mathcal{Y}_{\text{neg}}$).

**(b) Fabricating high-risk chest abnormalities:** Here, the adversary aims to potentially trigger unnecessary biopsies or follow-up examinations by inserting particular high-risk abnormalities. In this case, we sample the target text from the set of positive sentences in the ground-truth reports, which indicate the presence of critical abnormality ($y_{\text{target}} \in \mathcal{Y}_{\text{pos}}$).

### 3.2.2. Multimodal Adversarial Optimization

To fool the MLLM to generate an adversarial text report based on multimodal input, we begin by randomly initializing learnable visual perturbations $\delta_{ct} \in \mathbb{R}^{C \times H \times W \times D}$ and text prompt perturbations $\delta_t \in \mathbb{R}^{L \times d}$. $C$, $H$, $W$, and $D$ represents channel, height, width and depth dimension of 3D CT volume. $L$ and $d$ denote the length and hidden dimension of the text embedding. $\delta_{ct}$ is added to the clean 3D CT scan $x_{ct} \in \mathbb{R}^{C \times H \times W \times D}$ and $\delta_t$ to the text-prompt embedding $E_t \in \mathbb{R}^{L \times d}$ derived from text prompt $x_t$. We also localize the perturbation to the specific anatomical region using segmentation masks to isolate that region within the CT volume. This enables our proposed adversary to steer the adversarial optimization toward the loss associated with the corresponding anatomical subsection of the report. More specifically, let $M^{(r)} \in \{0,1\}^{H \times W \times D}$ be the binary segmentation mask for anatomical region $r$ (*e.g.,* lung), and let $\delta_{ct}$ be the global image perturbation. The perturbation applied to region $r$ is $\delta_{ct}^{(r)} = M^{(r)} \odot (\delta_{ct} + x_{ct})$. After the perturbed multimodal input is passed through the MLLM, we can then obtain the text report loss $\mathcal{L}$ and subsequently the perturbation's gradient corresponding to the image and text embedding $\nabla_{\delta_{ct}}$ and $\nabla_{\delta_t}$, respectively. The image- and text-prompt perturbation gradients are updated via gradient descent, followed by projection onto the allowable perturbation range. Figure 7 in Appendix E illustrates the evolution of adversarially optimized CT images across different optimization steps.

We update perturbations $\delta_{ct} \leftarrow \text{clip}(\delta_{ct} - \alpha_{ct} \, \text{sign}(\nabla_{\delta_{ct}}))$ and $\delta_t \leftarrow \text{clip}(\delta_t - \alpha_t \, \text{sign}(\nabla_{\delta_t}))$ using PGD (Madry et al., 2017). $\alpha_{ct}$ and $\alpha_t$ denote the step sizes for image and text embedding perturbation updates. To keep perturbations minimal while adversarially effective, we use PGD with adaptive early stopping (Li et al., 2025) where optimization stops once the generated report is sufficiently aligned with the target ($\geq \tau$ similarity by a text similarity metric) (Zhang et al., 2020). In our experiments, we use 1/255, 0.01, 16/255, 0.1, and 0.85 for $\alpha_{ct}$, $\alpha_t$, $\epsilon_{ct}$, $\epsilon_t$, and $\tau$, respectively. We set 100 as the maximum number of optimization steps.

## 4. Experimental Settings

### 4.1. Victim CT-to-text Report Generation Model

We evaluate our attack on two state-of-the-art 3D CT–to–text report generation models, Reg2RG (Chen et al., 2025) and CT-CHAT (Hamamci et al., 2024). Reg2RG (Chen et al., 2025) is a multimodal model composed of a 3D Vision Transformer (3D-ViT) (Wu et al., 2025) for volumetric CT encoding and a LLaMA2-7B decoder (Touvron et al., 2023). It incorporates anatomically grounded region tokens to improve alignment between visual features and organ-specific descriptions, enabling the generation of structured chest CT reports. CT-CHAT (Hamamci et al., 2024) combines a 3D CT-CLIP encoder with a LLaMA-3.1-8B decoder via a multimodal projector, leveraging CT-CLIP's contrastively learned 3D representations. All experiments use the official pretrained weights and inference pipeline released by the authors. [1],[2]

---

1. https://github.com/zhi-xuan-chen/Reg2RG
2. https://github.com/ibrahimethemhamamci/CT-CHAT

Table 1: Evaluation of adversarial attack on targeted organ and entire report. We evaluate our attack for both Targeted Suppression (Omission) and Targeted Fabrication (Insertion) of clinical findings. A higher ROUGE-L score indicates stronger adversarial effectiveness.

| Method | Organ-level | | Report-level | |
|---|---|---|---|---|
| | Suppression | Fabrication | Suppression | Fabrication |
| Reg2RG (Chen et al., 2025) (w/o attack) | 0.439 | 0.198 | 0.370 | 0.280 |
| CRA-RG (Reg2RG) | **0.937** | **0.888** | **0.410** | **0.294** |
| CT-CHAT (Hamamci et al., 2024) (w/o attack) | 0.445 | 0.513 | 0.281 | 0.276 |
| CRA-RG (CT-CHAT) | **1.000** | **1.000** | **0.761** | **0.508** |

## 4.2. Implementation and Benchmark Details

We evaluate our attack on the RadGenome-Chest dataset (Zhang et al., 2025b), using the 1,500 CT–report pairs from its validation split. All experiments are conducted on a single NVIDIA RTX A6000 (48 GB). As shown in Figure 2, both the volume encoder and language decoder remain frozen during attack optimization. For demonstration, we focus on adversarial manipulation of critical findings in the lung and breast.

## 4.3. Evaluation Setup for Adversarial CT Reports

We evaluate two targeted attack scenarios to determine whether the adversary can fool the MLLM to omit true findings and fabricate false ones. For each input CT scan, a target is randomly selected from a set of predefined clinical statements. Although the attack focuses on specific anatomies (the lung and breast), perturbations are applied to the entire CT volume and may affect other findings. To analyze these effects, we evaluate adversarial outcomes at both the organ and report levels. Organ-level evaluation measures how closely the generated organ-specific captions match the predefined targets. Report-level evaluation compares the full adversarial report to two idealized adversarial versions — one with all critical findings suppressed and one with all findings fabricated. For reference, we also compute the corresponding similarities for benign predictions to establish a baseline indicating how unlikely such manipulations are in the absence of an attack.

## 5. Experimental Results

### 5.1. Attack Success Rate of Targeted CT Report Manipulation

To assess whether the adversarial perturbations applied to multimodal input can lead to successful fooling of the MLLM in manipulating the critical findings, we compute ROUGE-L (Lin, 2004) similarity between the adversarially generated reports and predefined target texts as summarized in Table 1. Both Reg2RG and CT-CHAT show low organ- and report-level similarity with the adversarial target texts in the absence of an attack, indicating that the target descriptions rarely occur in normal report generation. In contrast, our attack substantially increases ROUGE-L scores for both targeted suppression and fabrication, demonstrating effective manipulation of localized clinical findings across two state-of-the-art CT report generation models. We observe consistently higher success at the organ level than at the report level, suggesting that localized findings are easier to manipulate due to their direct correspondence with visual features, whereas full reports require maintaining

Table 2: NLG-based Evaluation of Stealthiness of Adversarial Reports.

| Method | BL-1 | BL-2 | BL-3 | BL-4 | MTR | RG-L |
|---|---|---|---|---|---|---|
| Reg2RG (Chen et al., 2025) (w/o attack) | 0.473 | 0.365 | 0.296 | 0.249 | 0.441 | 0.367 |
| CRA-RG (Reg2RG) | 0.413 | 0.302 | 0.229 | 0.179 | 0.387 | 0.306 |
| CT-CHAT (Hamamci et al., 2024) (w/o attack) | 0.366 | 0.260 | 0.195 | 0.154 | 0.275 | 0.199 |
| CRA-RG (CT-CHAT) | 0.252 | 0.197 | 0.167 | 0.149 | 0.319 | 0.188 |

global clinical and linguistic consistency. While steering an entire report toward a fixed adversarial target remains challenging, these results highlight a critical vulnerability whereby clinically salient findings can be selectively inserted or removed without fully compromising global report coherence. Qualitative examples of ground-truth and adversarial CT reports are provided in Figure 4 and Figure 5 in Appendix A. We analyze attack robustness under varying perturbation budgets in Table 7 (Appendix B), present qualitative comparisons between clean and adversarial CT images with difference maps in Figure 6 (Appendix C), and report quantitative quality metrics for adversarial CT images in Table 8 (Appendix D).

### 5.2. Stealthiness of Adversarial CT Reports

To assess whether adversarial CT reports generated by attacking the MLLM remain stealthy and inconspicuous, we quantify the extent to which they preserve the structural characteristics of reports produced from clean inputs. Specifically, we compute Natural Language Generation (NLG) metrics between adversarial and ground-truth reports and compare these scores with those obtained by evaluating benign reports[3] against their corresponding ground-truth texts. In this setting, NLG metrics serve as proxies for structural similarity because they indicate how closely the adversarial reports retain the overall textual organization of the original reports. We compute BLEU-n (Papineni et al., 2002) (n-gram overlap), ROUGE-L (Lin, 2004) (longest-common-subsequence similarity), and METEOR (Lavie and Agarwal, 2007) (semantically informed matching), as summarized in Table 2. As shown in the Table 2, adversarial reports exhibit modest reductions in NLG scores relative to benign baselines; however, the decline is not substantial given that these reports are generated from adversarially perturbed CT scans and prompt embeddings. We observe a relatively larger drop in NLG scores for CT-CHAT under attack compared to Reg2RG. We hypothesize that this is largely due to CT-CHAT's higher fooling rate (as shown in Table 1), which allows adversarial perturbations to more strongly manipulate the reports. Despite successfully suppressing critical clinical findings, the adversarial CT reports largely preserve the structural form of the original CT reports and therefore remain stealthy.

### 5.3. Transferability of Adversarial CT Perturbations

To analyze if our induced CT perturbations persist across different MLLMs, we conduct black-box transferability experiments. We evaluate black-box transferability by applying adversarial CT volumes optimized against the Reg2RG model with a LLaMA-2-7B decoder (Chen et al., 2025) to a target model employing the same 3D-ViT encoder but a different

---

3. Benign CT reports denote reports generated from clean, unattacked inputs.

Table 3: Transferability of adversarial CT perturbations across language decoders: Adversarial CT volumes generated with Reg2RG (3D ViT + LLaMA2-7B) are evaluated for adversarial report generation effectiveness on a Mistral-7B decoder. We report average attack success rate on organ level and report level.

| Method | Organ-level | Report-level |
|---|---|---|
| Reg2RG (LLaMA2-7B) (Chen et al., 2025) (w/o attack) | 0.319 | 0.325 |
| Reg2RG (Mistral-7B) (Chen et al., 2025) (w/o attack) | 0.263 | 0.404 |
| CRA-RG (LLaMA2-7B) | 0.913 | 0.352 |
| CRA-RG (Mistral-7B) | 0.795 | 0.395 |
| CRA-RG (LLaMA2-7B → Mistral-7B) | 0.594 | 0.284 |
| CRA-RG (Mistral-7B → LLaMA2-7B) | 0.387 | 0.334 |

Table 4: Organ Recognition Performance Under Multimodal Adversarial Attack: We examine the ability of MLLM to identify the organs in adversarial CT scans. A decrease in Recall and F1-score indicates the strength of our attack in deteriorating the MLLM's organ detection ability.

| Organ | Benign Reports | | Adversarial Reports | | Difference | |
|---|---|---|---|---|---|---|
| | Recall | F1 | Recall | F1 | Δ Recall | Δ F1 |
| Abdomen | 0.997 | 0.997 | 0.649 | 0.733 | 0.348 | 0.264 |
| Bone | 0.999 | 0.999 | 0.652 | 0.785 | 0.347 | 0.214 |
| Breast | 0.967 | 0.967 | 0.908 | 0.863 | 0.059 | 0.104 |
| Esophagus | 0.999 | 0.999 | 0.941 | 0.965 | 0.058 | 0.034 |
| Heart | 0.995 | 0.995 | 0.938 | 0.947 | 0.057 | 0.048 |
| Lung | 0.807 | 0.807 | 0.008 | 0.009 | **0.799** | **0.798** |
| Mediastinum | 0.995 | 0.993 | 0.946 | 0.850 | 0.049 | 0.143 |
| Pleura | 0.807 | 0.807 | 0.193 | 0.319 | 0.614 | 0.488 |
| Thyroid | 0.962 | 0.975 | 0.894 | 0.785 | 0.068 | 0.190 |
| Trachea & Bronchi | 0.973 | 0.977 | 0.930 | 0.937 | 0.043 | 0.040 |
| Average | 0.950 | 0.952 | 0.706 | 0.719 | 0.244 | 0.233 |

language decoder (Mistral-7B). [4] During attack optimization, gradients flow through the entire pipeline, but the perturbation ultimately manipulates the visual and textual representations, not the visual encoder or language decoder parameters. As shown in Table 3, perturbations optimized against Reg2RG (LLaMA-2-7B) demonstrate significant transferability to a Mistral-7B target, with organ-level ASR reaching 0.594, which is nearly double the clean baseline (0.319). In contrast, adversarial transfer from Mistral-7B to LLaMA2-7B is weaker at the organ level, and report-level transferability in this direction (0.334) falls below the corresponding clean baseline (0.404). This suggests that while the attack can effectively degrade the correctness of generated reports under black-box transfer, it is less reliable at inducing specific targeted clinical descriptions. While white-box efficacy remains highest (0.913), the success of these cross-decoder attacks suggests that the vulnerability is not specific to a particular decoder architecture but rather arises from shared multimodal alignment mechanisms and common generation dynamics across LLM decoders. These findings underscore the practical risk of our threat model that an attacker possessing white-box knowledge of a commonly used medical vision encoder can effectively compromise a black-box system even if the specific language model remains proprietary or unknown.

---

4. Due to backbone-specific volumetric preprocessing, we assess black-box transfer by fixing the 3D-ViT encoder and varying the language decoder.

Table 5: Effect of Adversarial Attack on Detection of Chest Abnormalities: This table reports the degradation in detection of chest findings from CT reports under our proposed adversarial threat. The decrease in Recall and F1 indicate the effectiveness of our attack at concealing critical abnormalities in the generated reports.

| CT findings | Benign Reports | | Adversarial Reports | | Difference | |
|---|---|---|---|---|---|---|
| | Recall | F1 | Recall | F1 | Δ Recall | Δ F1 |
| Arterial wall calcification | 0.62 | 0.62 | 0.31 | 0.43 | 0.31 | 0.19 |
| Cardiomegaly | 0.12 | 0.16 | 0.0 | 0.0 | 0.12 | 0.16 |
| Coronary artery wall calcification | 0.61 | 0.55 | 0.18 | 0.27 | 0.43 | 0.28 |
| Emphysema | 0.22 | 0.26 | 0.01 | 0.03 | 0.21 | 0.23 |
| Atelectasis | 0.65 | 0.39 | 0.0 | 0.0 | 0.65 | 0.39 |
| Lung opacity | 0.24 | 0.38 | 0.03 | 0.06 | 0.21 | 0.32 |
| Pulmonary fibrotic sequela | 0.11 | 0.17 | 0.0 | 0.0 | 0.11 | 0.17 |
| Pleural effusion | 0.38 | 0.50 | 0.03 | 0.05 | 0.35 | 0.45 |
| Consolidation | 0.24 | 0.31 | 0.0 | 0.0 | 0.24 | 0.31 |
| Average | 0.35 | 0.37 | 0.06 | 0.09 | 0.29 | 0.28 |

## 5.4. Thoracic Organs Recognition after Adversarial Attack on MLLM

The baseline MLLM we attack can both identify thoracic organs and generate full CT reports (Chen et al., 2025). Through prompting, the model is instructed to both recognize anatomical structures and produce a comprehensive description of the scan. To evaluate how our adversarial perturbations affect the MLLM's ability to detect thoracic organs, we assess organ-recognition performance under clean and adversarial conditions using Recall and F1 scores. The organ recognition performance is shown in Table 4. The baseline model recognizes most organs with near-perfect Recall and F1 scores. With adversarially manipulated multimodal inputs, recognition performance decreases across nearly all organs, indicating that adversarial perturbations disrupt the model's grounding ability. The most severe degradation occurs for the lung, where Recall and F1 drop sharply from 0.807 to nearly zero, consistent with our attack targeting lung-related findings. In contrast, regions less directly tied to the adversarial objective (*e.g.,* Heart, Esophagus, Trachea & Bronchi) retain relatively high performance, though still lower than the benign baseline. Overall, these results indicate that adversarial perturbations not only manipulate targeted clinical findings but also impair the MLLM's broader ability to correctly identify anatomical thoracic regions.

## 5.5. Impact of Adversarial Attack on Detection of Clinical Findings

To evaluate how our attack affects the detection of clinically meaningful findings that are extracted from generated chest CT reports, we employ Clinical Efficacy (CE) metrics (Chen et al., 2020). In contrast to conventional NLG metrics that assess textual similarity, CE metrics measure diagnostic fidelity by determining whether key clinical abnormalities are correctly captured in the generated text. Because our adversary aims to attack the MLLM to alter critical diagnostic content, CE metrics provide a direct assessment of the attack's clinical consequences. We use the RadBERT text classifier (Yan et al., 2022) to automatically extract abnormalities from reports, and quantify performance using Recall and F1 score. Table 5 summarizes the results. For each abnormality, we present Recall and F1 scores derived from benign and adversarial reports, measured against the abnormalities

Table 6: Ablation analysis of modality-specific and multimodal adversarial attacks for targeted suppression and fabrication at the organ and report levels. Higher ROUGE-L scores denote stronger attack effectiveness.

| Method | Organ-level | | Report-level | |
|---|---|---|---|---|
| | Suppression | Fabrication | Suppression | Fabrication |
| Baseline (Chen et al., 2025) (w/o attack) | 0.439 | 0.198 | 0.370 | 0.280 |
| CRA-RG (visual) | 0.922 | 0.887 | **0.456** | 0.272 |
| CRA-RG (text) | 0.518 | 0.719 | 0.346 | 0.278 |
| CRA-RG (visual + text) | **0.937** | **0.888** | 0.410 | **0.294** |

mentioned in the ground-truth report. We found that, in adversarial reports, the detection of critical chest findings declines sharply. Many abnormalities, including cardiomegaly, atelectasis, consolidation, and fibrotic sequela, become completely undetectable, indicating that the adversarial attack substantially disrupts the clinical information encoded in the generated reports and thus reduces their diagnostic reliability.

### 5.6. Ablation Study on the Impact of Attacking Individual Modalities

We perform an ablation analysis on Reg2RG baseline model (Chen et al., 2025) to evaluate the relative effectiveness of visual-only, text-only, and multimodal input perturbations in adversarial report generation. As shown in Table 6, visual perturbations achieve high anatomical organ-level success (0.92), while text-only perturbations are less effective, likely because the image remains unchanged. The strongest manipulation (0.937) occurs when both modalities are perturbed. Notably, fabrication is more difficult than suppression, suggesting that degrading visual evidence makes it harder for the model to justify the insertion of nonexistent findings.

### 5.7. Analysis of Multimodal Embedding Space under Adversarial Attack

To understand how multimodal adversarial inputs affect the MLLM's embedding space, we compute cosine similarity between ground-truth embeddings and embeddings from non-adversarial versus adversarial reports. Figure 3 shows the cosine similarity matrices between ground-truth embeddings and embeddings from non-adversarial (left) versus adversarial reports (right). The non-adversarial similarity matrix exhibits broad consistency with the ground-truth embeddings. In contrast, the adversarial matrix exhibits pronounced horizontal bands, indicating that ground-truth tokens map strongly to a narrow subset of adversarial embeddings. This pattern reflects a distortion in the embedding space induced by the adversarial perturbations, demonstrating that the attack not only alters the textual output but also reshapes internal semantic representations.

## 6. Discussion and Limitations

This work presents the first systematic study of adversarial attacks on large CT report generation MLLMs. By perturbing multimodal inputs, we generate clinically risky adversarial reports that include targeted suppression and fabrication of findings. While these are the focus of this study, other, more subtle adversarial forms may exist. Our threat model

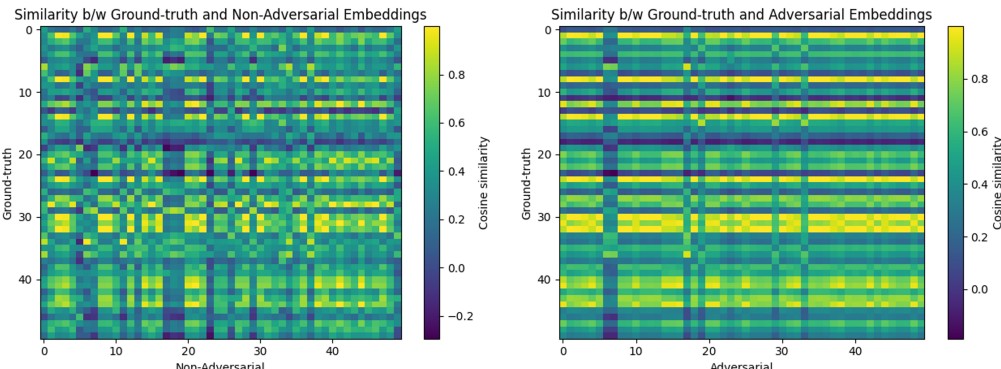

Figure 3: Cosine similarity matrices between ground-truth token embeddings and embeddings from non-adversarial (left) and adversarial (right) reports.

assumes a white-box adversary, which is realistic given the public availability of widely used medical vision encoders. Cross-decoder transferability results suggest that vulnerabilities lie in shared visual representations and multimodal alignment, meaning attacks may persist even with proprietary language models. Finally, developing defenses against such adversarial manipulation remains a key avenue for future work.

## 7. Conclusion

In this work, we presented Clinically Risky Adversarial Report Generation (CRA-RG), a new threat model that characterizes how chest CT radiology report generation systems can be adversarially manipulated to produce clinically dangerous outputs. Our threat model, CRA-RG focuses on clinically meaningful manipulations, including the omission and fabrication of critical findings such as nodules, consolidations, and pleural abnormalities. To instantiate this threat model, we introduced a multimodal targeted adversarial attack that jointly perturbs CT volumes and conditioning text embeddings, enabling fine-grained control over specific anatomical regions. Our experiments on the RadGenome 3D chest CT dataset demonstrated that state-of-the-art multimodal report generation models are highly susceptible to adversarial perturbations. These results provide the first empirical evidence that modern chest CT report generation systems can be driven to produce harmful clinical recommendations — including missing high-risk findings or fabricating nonexistent abnormalities — raising critical safety concerns for real-world deployment.

## Acknowledgments

This work was supported by the IITP (Institute of Information & Coummunications Technology Planning & Evaluation)-ITRC (Information Technology Research Center) grant funded by the Korea government (Ministry of Science and ICT) (IITP-2026-RS-2023-00258649, 50%), and by the National Research Foundation of Korea (NRF) grant funded by the Korea government (MSIT) (No. RS-2024-00334321, 50%). We also thank Minkuk Kim and Youngseob Won for helpful discussions in the early stages of this work.

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

# Appendix A. Example of Real and Adversarial CT Report

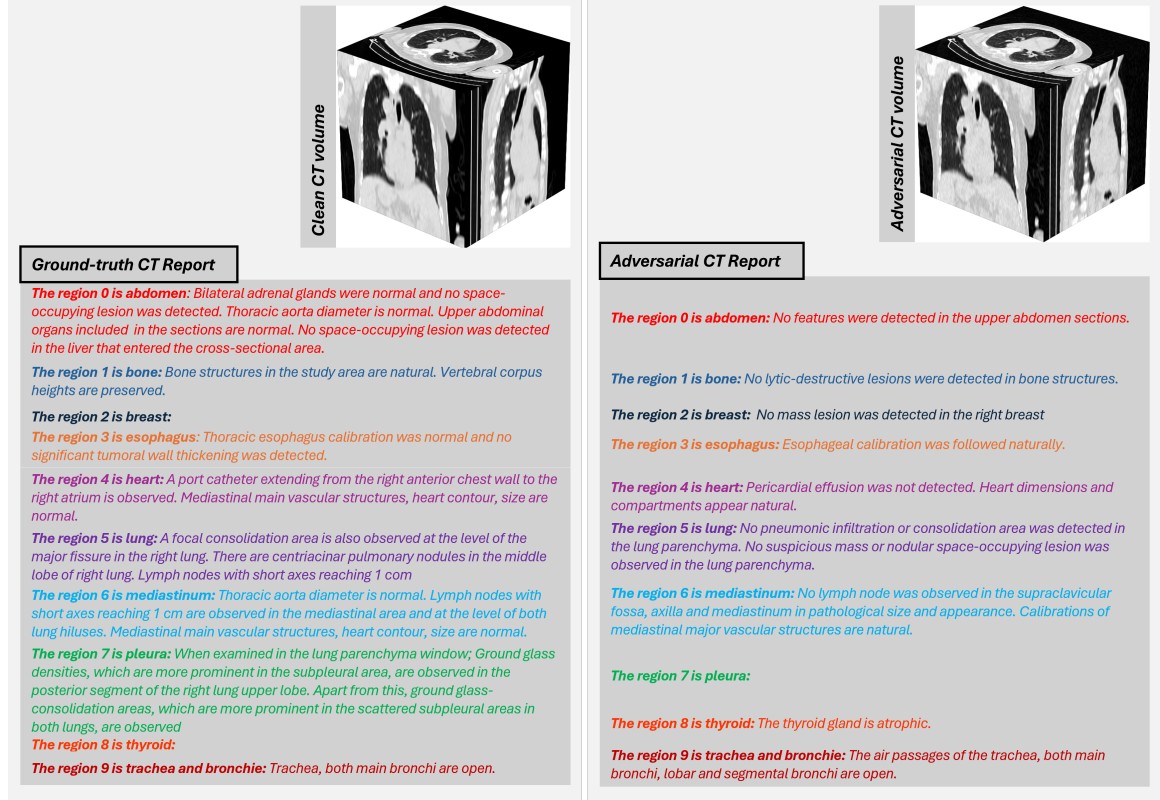

Figure 4: We present an example of a real CT report (left) and an adversarially generated CT report (right) using our proposed multimodal perturbations. Note that the CT report generation model has been successfully fooled to omit the majority of the significant clinical findings in Chest CT (e.g., Consolidation in lungs and lymph nodes in Mediastinum are successfully suppressed in Adversarial CT Report).

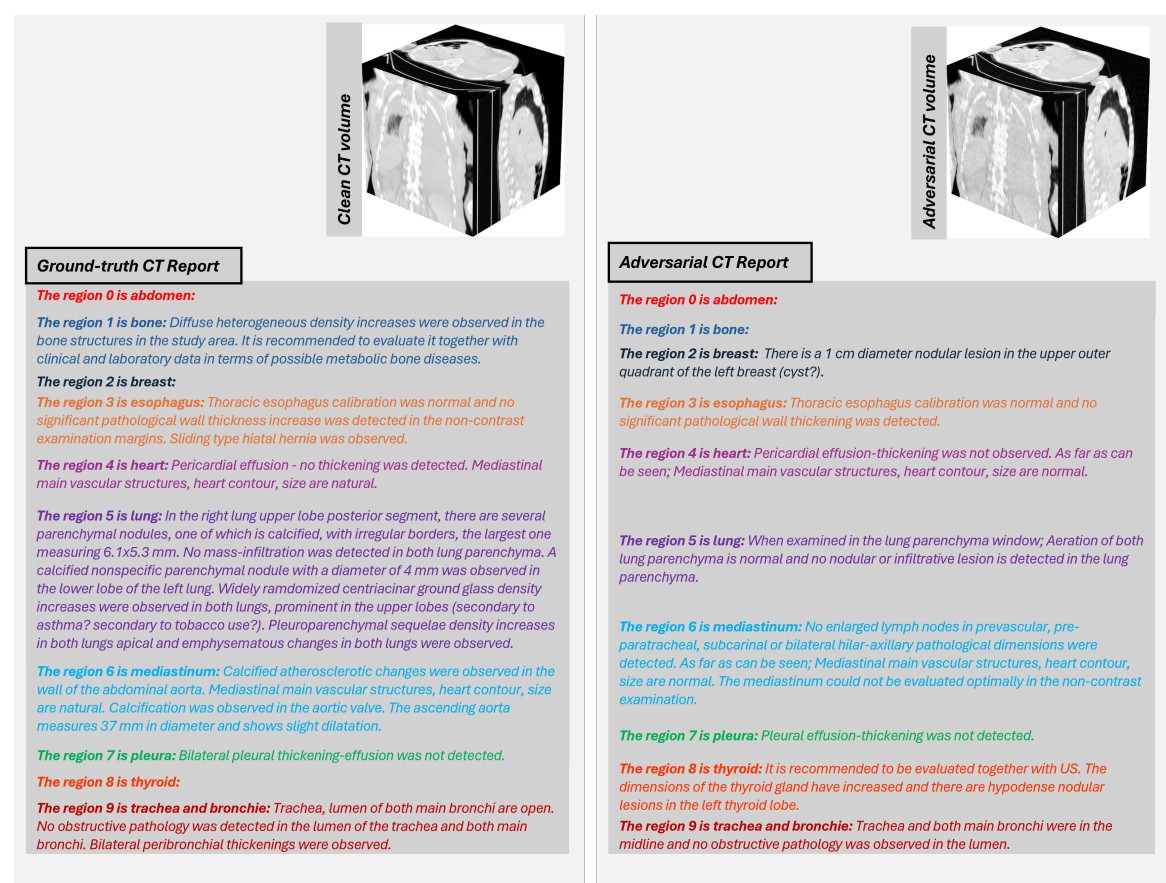

Figure 5: Another Example of a clean (left) and adversarial CT report (right) generated by attacking the MLLM through adversarial multimodal input. The abnormalities in the bone and lung have been successfully omitted.

## Appendix B. Ablative Study on Perturbation Budget

To study the impact of the CT perturbation budget ($\epsilon_{ct}$) on attack success, we perform an ablation experiment in which we vary $\epsilon_{ct}$ from 16/255 to 8/255 and 4/255 on CT volumes that are successfully attacked at $\epsilon_{ct} = 16/255$. We evaluate performance by reporting the attack failure rate.

Table 7 analyzes how the CT perturbation budget ($\epsilon_{ct}$) affects attack success. Reducing the perturbation budget to 8/255 results in a small increase in attack failures (1%), indicating that the attack remains largely effective even under tighter visual constraints. When ($\epsilon_{ct}$) is further reduced to 4/255, the attack failure rate increases to 8%, suggesting the emergence of a minimum effective perturbation threshold below which adversarial optimization becomes less reliable. Overall, these results highlight a trade-off between attack success and perturbation strength and demonstrate that the proposed attack remains effective at relatively low ($\epsilon_{ct}$) values.

Table 7: Attack Robustness Under Varying CT Perturbation Budgets

| Perturbation Budget $\epsilon_{ct}$ | Attack Failure Rate |
| --- | --- |
| 16/255 | 0.0 |
| 8/255 | 0.01 |
| 4/255 | 0.08 |

## Appendix C. Visual Comparison of Clean and Adversarial CT Scans

We present examples of clean CT images alongside their adversarial counterparts and corresponding difference maps in Figure 6. As illustrated, the adversarial perturbations are visually imperceptible and do not introduce noticeable artifacts, while the difference maps reveal subtle, spatially distributed changes that are sufficient to induce significant manipulation of the generated reports.

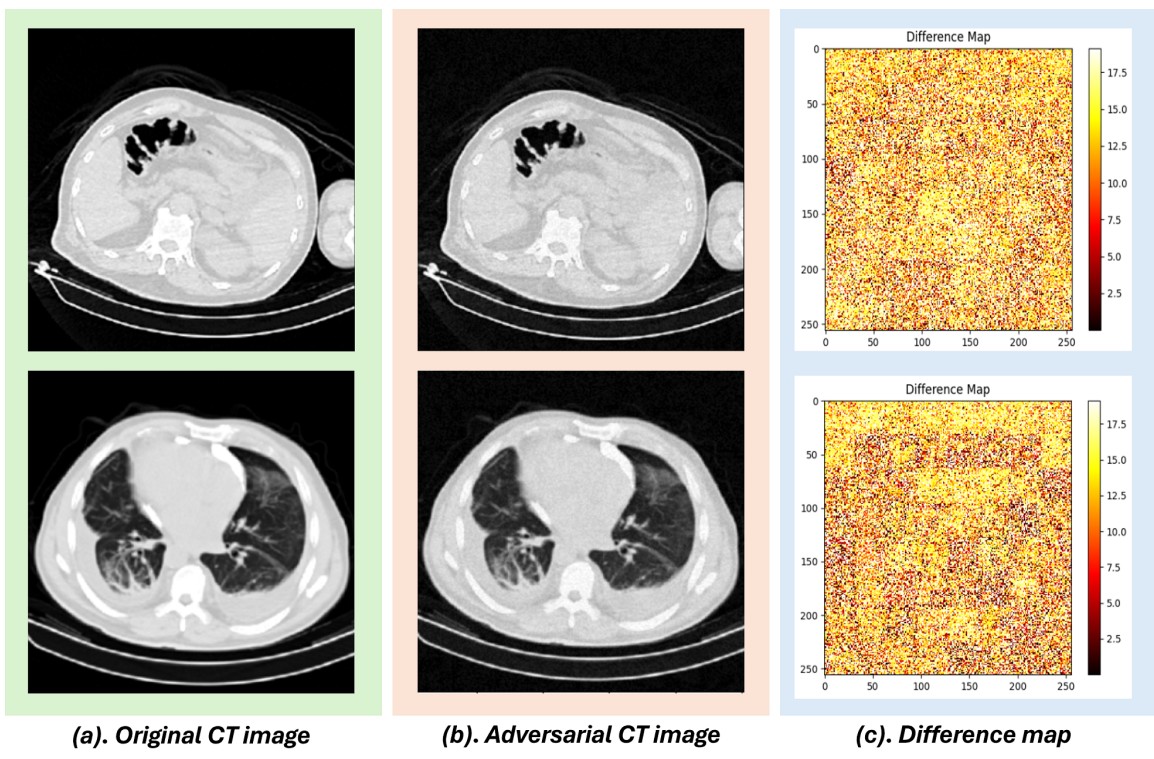

*(a). Original CT image*   *(b). Adversarial CT image*   *(c). Difference map*

Figure 6: Comparison of the Visual Quality of Original CT images, Adversarial images, and their difference maps.

## Appendix D. Quantifying the Stealth of Adversarial CT Perturbations

To assess the deviation of adversarial CT images from their original counterparts, we compute perceptual similarity metrics, including SSIM (Wang et al., 2004), as well as the mean and maximum voxel-wise intensity differences between clean and adversarial volumes measured in Hounsfield Units (HU). These metrics jointly quantify structural similarity and perturbation magnitude, providing insight into the visual stealth of the proposed attack. As shown in Table 8, the adversarial CT images achieve a high SSIM score of 93.27%, indicating strong structural preservation relative to the original scans. At the same time, the mean and maximum intensity differences remain limited (14.74 HU and 19.15 HU, respectively), suggesting that the perturbations introduce only subtle intensity variations that are difficult to perceive by visual inspection. Together, these results demonstrate that the proposed attack maintains high visual fidelity.

Table 8: Evaluation of Visual Adversarial Stealth

| Attack Method | Mean HU Diff. | Max. HU Diff. | SSIM (%) |
|---|---|---|---|
| CRA-RG | 14.74 | 19.15 | 93.27 |

## Appendix E. Adversarial Optimization of CT images

The proposed attack is initialized with random noise, which is then iteratively optimized under anatomical and perceptual constraints to manipulate the generated report. As a result, the perturbations in the early optimization steps indeed resemble unstructured noise, which is expected given the initialization. The final adversarial perturbations are not equivalent to fixed random noise. Through iterative optimization, the perturbations become task-driven and exploit model-specific vulnerabilities, resulting in consistent, targeted changes to the generated reports. We show the evolution of adversarial images at different steps in Figure 7.

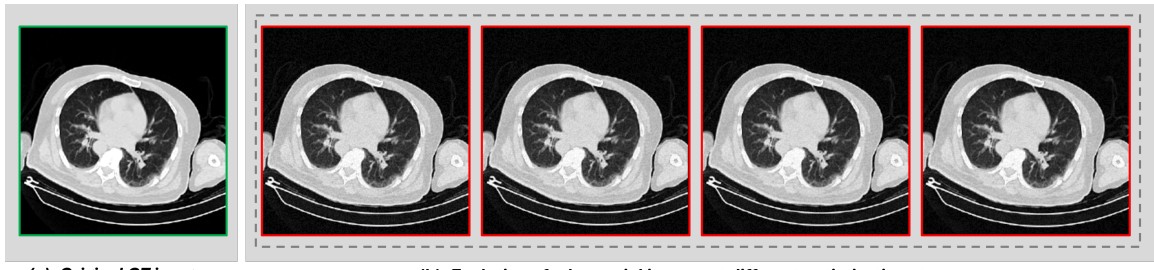

(a). Original CT image    (b). Evolution of adversarial images at different optimization steps

Figure 7: Evolution of Adversarial CT images against different optimization steps.

