# OpenReview forum: "No Evidence of Disease: Clinically-Risky Adversarial Chest CT Report Generation"
_MIDL.io/2026/Conference — MIDL 2026 Poster_

### Official Review · Reviewer_VJoT · 2025-12-23

**Confidence:** 3
**Preliminary Rating:** 2
**Final Rating:** 3

**Summary:**

The authors propose an evasion attack framework targeting a CT-based multimodal large language model (MLLM). The method assumes direct access to the victim model and identifies perturbations in both image and text inputs that cause the model to suppress clinically relevant information or hallucinate diagnoses.

**Strengths:**

The strengths of the paper are :

1. The method is generally easy to follow and understand, and the paper is reasonably well written and explained, though some sections could benefit from additional clarity.

2. Detecting and identifying vulnerabilities in current medical multimodal large language models (MLLMs) is a valid and important area of research.

**Weaknesses:**

There are some major weaknesses in my opinion :

1. **Novelty:** Methodology is easy, but not novel. Similar methodologies already exist in the literature. The novelty of the manuscript lies in MLLM usage for CT dataset.

2. **Problems with Validation:** The manuscript does not provide any validation to ensure that the adversarial perturbations do not inadvertently alter the underlying anatomical structures in the images. If manual evaluation of the perturbed images also changes, this would suggest that the adversarial modifications are causing actual changes in the anatomy (which can be inferred from table 2 performance degradation as well), rather than merely exposing model vulnerabilities. This limitation is not discussed.

- Furthermore, no qualitative results are presented, such as visualizations of the perturbations $\delta_{ct}$. If the perturbations resemble simple noise, a comparison with standard noise-based perturbations might yield similar outcomes, raising questions about the utility of the proposed method.

- Additionally, from Figure 2, it appears that a single noise generator is applied to the entire dataset, rather than generating sample-dependent perturbations. It is unclear whether the pipeline is optimized per sample, which would result in sample-specific perturbations, or if it is dataset-wide. Clarification and discussion of this aspect are needed.

- An Analysis of what $\delta_{ct}$ for different Adverserial goal in section 3.2, is missing which i believe is essential.

3. **Access to Victim Model:** For medical image, access of victim model is a big assumption. Because if the model is trained on local or private datasets, that model might not be released to public. So this assumption might not be a valid assumption. If it is, then authors should expand more on this and justify their choice with specific examples and applications.

**Detailed Comments:**

Qualitative results with $\delta_{ct}$ for images are essential for analysis. For other comments refer to Weaknesses section.

**Justification Of Final Rating:**

With the new visualizations added to the paper, I believe it has improved; however, I am not fully convinced of its novelty. While the application to CT is novel, the methodology itself is not. Therefore, I have decided to increased my score.

**Justification Of The Preliminary Rating:**

The methods presented in the manuscript lack novelty and do not provide sufficient analysis to warrant acceptance in their current form. Numerous questions remain regarding the validity of the results, and the absence of qualitative analysis further suggests that additional work is required before this study can be considered for publication.

**Questions To Address In The Rebuttal:**

The authors should prioritize addressing Points 2 and 3 in the weaknesses section. While Point 1 is less critical and can be disregarded, my rating is primarily based on the issues raised in the other two points.

---

> ### Author Response · Authors · 2026-01-25
> **Official Comment by Authors**
>
> We sincerely thank the reviewer for their comments and for giving us an opportunity to express our work with more clarity.
>
> **1. Perceptual Evaluation:**
>
> We have added a new section in **Appendix C (Figure 6)** and **Appendix D (Table 8)** to our revised paper, showing the original CT images, adversarial CT images, and difference maps. Please see the relevant sections of our revised paper. Figure 6 and Table 8 show that our attack induces subtle perturbations without causing any anatomical changes.
>
> **2. Simple noise and adversarial perturbations:**
>
> Adversarial perturbations are optimized with respect to the loss, so they do not resemble simple noise. We would like to clarify that the proposed attack is initialized with random noise, which is then iteratively optimized under anatomical and perceptual constraints to manipulate the generated report. As a result, the perturbations in the early optimization steps indeed resemble unstructured noise, which is expected given the initialization. Crucially, however, the final adversarial perturbations are not equivalent to fixed random noise. Through iterative optimization, the perturbations become task-driven and exploit model-specific vulnerabilities, resulting in consistent, targeted changes to the generated reports.
>
> To address the reviewer's concern, we have added a section in **Appendix E (Figure 7)** to show the evolution of Adversarial CT images.
>
> **3. Sample dependent perturbations:**
>
> We would like to clarify that our attacking framework is sample-dependent since the perturbations are optimized for each CT volume independently.
>
> **4. Access to Victim Model:**
>
> We assume white-box settings for our attack. However, in our revised paper, we have added black-box transferability experiments (Section 5.3 and Table 3).
>
> **5. Novelty:**
>
> We thank the reviewer for this comment. The primary novelty of this work lies in formulating a previously unexplored problem, **adversarial CT report generation**, and demonstrating how multimodal inputs (3D CT volumes and textual prompts) can be systematically manipulated to induce clinically meaningful report-level errors.
>
> To our knowledge, this is the first work to explicitly define, formalize, and evaluate adversarial attacks on radiology report generation models. This includes (i) defining a threat model tailored to multimodal medical report generation, (ii) proposing clinically grounded perturbations on volumetric CT data, and (iii) introducing evaluation protocols that assess adversarial effects at the organ and report level rather than solely on classification outputs.
>
> While the optimization procedure is easy to understand, its application to CT report generation, the multimodal attack formulation, and the associated clinical interpretation represent the key novel contributions of this paper.

---

> ### Author Response · Authors · 2026-01-29
>
> As the discussion period nears its end, we would like to follow up on our earlier reminder. We have submitted a revised manuscript and responses addressing all comments. Please let us know if any further clarification is needed.
>
> Thank you again for your time and attention.

---

> ### Comment · Area_Chair_bGNM · 2026-01-31
> **Rebuttal discussion (optional)**
>
> Thanks again for your reviews. Discussion is a unique feature of MIDL that helps foster constructive engagement in the evaluation process, so if you have time, please feel free to look at the authors’ rebuttal, join the discussion, and update your final rating as you see fit. Thanks.

---

### Official Review · Reviewer_uhyw · 2026-01-09

**Confidence:** 2
**Preliminary Rating:** 4
**Final Rating:** 4

**Summary:**

This paper studies adversarial vulnerabilities of multimodal chest CT report generation models, focusing on clinically dangerous failures rather than generic text corruption. The authors introduce Clinically-Risky Adversarial Report Generation (CRA-RG), a threat model targeting realistic but harmful manipulations of CT reports, specifically the suppression or fabrication of high-risk findings (e.g., lung nodules, consolidation). To instantiate this threat, they propose a targeted multimodal white-box attack that jointly perturbs 3D CT volumes (optionally localized to anatomical regions) and text-prompt embeddings using PGD. Experiments on the RadGenome chest CT dataset show that a state-of-the-art CT-to-text MLLM can be reliably induced to omit or hallucinate clinically meaningful findings while preserving report fluency and structure, highlighting a serious safety risk for automated radiology reporting systems

**Strengths:**

- Clear and clinically grounded threat model
The paper goes beyond generic “misleading outputs” and precisely defines what constitutes a clinically risky adversarial report (omission or fabrication of critical findings), which is highly appropriate for medical AI safety.

- Strong empirical demonstration of harm
The attack is shown to substantially reduce recall and F1 for both organ recognition (Table 3) and abnormality detection (Table 4), including near-complete suppression of lung findings, which convincingly demonstrates clinical risk.

- Multimodal and anatomy-aware attack design
Jointly perturbing CT volumes and prompt embeddings, with optional region-specific masking, is a realistic and technically well-motivated attack strategy for MLLM-based report generation systems.

- Comprehensive evaluation beyond text similarity
The use of organ-level metrics, clinical efficacy metrics, and embedding-space analysis strengthens the argument that the attack affects semantic grounding and not just surface-level text.

- Well written and well contextualized
The paper is clearly structured, technically precise, and engages appropriately with prior work on MLLMs, adversarial attacks, and radiology report generation.

**Weaknesses:**

- Strong white-box threat model limits realism
The attack assumes full access to model parameters and gradients, which may overestimate risk relative to many real-world deployment scenarios.

- Perturbation perceptibility is not thoroughly assessed
While perturbation bounds are specified, the paper does not provide quantitative or human-perceptual analysis of whether CT perturbations remain clinically imperceptible to radiologists.

- Reliance on automatic text and clinical extractors
Clinical impact is evaluated using automated classifiers (e.g., RadBERT), and no human expert validation is provided to confirm that adversarial reports would plausibly mislead clinicians.

- Limited model diversity
All experiments are conducted on a single CT-to-text architecture (Reg2RG), leaving open questions about generalization to other report generation models or training paradigms.

**Detailed Comments:**

- Clarify the perceptual and clinical imperceptibility of the CT perturbations, for example by reporting intensity statistics, visual difference maps, or expert radiologist feedback.

- Discuss how the proposed attack might transfer to more realistic settings with partial or black-box access, or to other CT report generation architectures beyond Reg2RG.

- Provide additional details on how anatomical masks are obtained and whether segmentation errors affect the effectiveness or localization of the attack.

- Include a brief discussion on potential mitigation strategies or defenses to help contextualize the findings within a broader safety framework.

**Justification Of Final Rating:**

The authors' rebuttal address most of my concerns. Only concern remains is human evaluation, which authors did not address directly but mentioned in future work. I will keep my original positive score. However, my confidence score is still low as I am not an expert in this field.

**Justification Of The Preliminary Rating:**

I assign a positive rating to this paper because it identifies and rigorously demonstrates a clinically meaningful and previously underexplored safety risk in multimodal chest CT report generation systems. The proposed Clinically Risky Adversarial Report Generation (CRA-RG) threat model is well motivated and clearly defined, focusing on realistic failure modes such as the omission or fabrication of high-risk findings rather than generic text corruption.

**Questions To Address In The Rebuttal:**

- How perceptually and clinically imperceptible are the proposed CT perturbations in practice, and can this be supported by quantitative measures or expert (radiologist) assessment?

- To what extent do the observed vulnerabilities generalize to other CT report generation models or to settings with limited or black-box access to the model?

- How sensitive is the attack to the quality and accuracy of the anatomical segmentation masks used for region-specific perturbations?

- Do the authors have preliminary insights into potential defenses or training strategies that could mitigate the clinically risky adversarial behaviors identified in this work?

---

> ### Author Response · Authors · 2026-01-25
> **Official Comment by Authors**
>
> We sincerely thank the reviewer for their review of our work and for providing thoughtful insights.
>
> **1. White-box Model access:**
>
> We thank the reviewer for the thoughtful feedback regarding the threat model. We would like to clarify that our white-box approach is intended to expose fundamental risks in 3D Medical MLLMs. Our intent in adopting a white-box setting is to isolate and characterize the vulnerability of the report-generation pipeline itself, independent of confounding factors such as query limits or partial access. Our threat model considers a white-box but write-restricted adversary who has access to the model architecture and parameters (e.g., in a research or enterprise environment) and to patient CT data, but cannot modify model weights or post-hoc outputs due to integrity checks. In such realistic insider scenarios, the only feasible attack surface is the visual input and intermediate embeddings, motivating the study of bounded adversarial perturbations rather than direct output tampering. Furthermore, our framework has significant practical utility for Privacy-Preserving. By using our method, patients or data owners can apply **'adversarial cloaks'** to their CT images. These perturbations prevent unauthorized VLMs from extracting sensitive diagnostic information.
>
> **2. Perturbation perceptibility:**
>
> We assess adversarial image drift using qualitative comparisons with difference maps in **Figure 6 (Appendix C)** and **quantitative perceptual metrics (SSIM and mean/max HU differences)** in **Table 8 (Appendix D)**.
>
> **3. Human expert validation:**
>
> We acknowledge that human expert validation would provide stronger clinical evidence. However, due to feasibility constraints, we adopt widely used automated clinical extractors (e.g., RadBERT) as proxy measures, which are commonly employed in large-scale medical AI evaluations. We explicitly note this limitation and leave human-in-the-loop validation for future work.
>
> **4. Generalization to multiple CT report generation models:**
>
> To address the concern about generalization beyond a single model, we extended our evaluation to **CT-CHAT** (Hamamci2024). Unlike our original baseline (Reg2RG), CT-Chat utilizes a fundamentally different architecture, including a LLaMA-3.1 8B language decoder and a CT-CLIP based visual encoder. We added the CT-CHAT model description in **Section 4.1** and the results are given in **Section 5.1** and **Table 1**. We have also evaluated the effect on NLG metrics when the CT-CHAT model is attacked to generate adversarial CT reports. The results for **NLG metrics** are shown in **Table 2**. The success of our attack on two state-of-the-art CT reporting MLLMs, which utilize different encoders and decoders, demonstrates that the vulnerability is model-agnostic and persists across modern 3D medical VLM designs.
>
> **5. Black-box transferability:**
>
> We evaluate black-box transferability by transferring adversarial CT volumes from **Reg2RG (LLaMA-2-7B)** to a model with the same 3D-ViT encoder and a different decoder **(Mistral-7B)**, with results reported in **Section 5.3** and **Table 3**.
>
> Transfer success rates of 59.4% and 38.7% demonstrate that white-box access to a shared vision encoder can compromise black-box systems with unknown language decoders.
>
> **6. Nosiy segmentation masks:**
>
> The anatomical masks used in our experiments are obtained from pre-existing organ segmentation annotations provided with the dataset (CT-RATE). These masks are not are not optimized jointly with the attack.
>
> To assess the sensitivity of the proposed attack to segmentation inaccuracies, we conducted an additional experiment in which we intentionally introduced noise into the anatomical masks by applying morphological dilation over several voxels and re-ran the attack using these noisy masks. Our results indicate that the attack remains effective under such mask perturbations. This suggests that moderate segmentation inaccuracies do not critically affect either the effectiveness of the attack.
>
> In this initial robustness analysis, we focused on mask dilation and did not exhaustively evaluate other types of segmentation errors (e.g., erosion). We will clarify this limitation in the camera-ready version and leave a more comprehensive analysis of segmentation robustness to future work.
>
> **7. Defense Strategies:**
>
> The primary goal of this work is to identify and characterize previously unexplored vulnerabilities in radiology report generation models. Our findings suggest several promising directions for mitigating clinically risky adversarial behaviors. These include adversarial or robustness-aware training of the report generation model, incorporating consistency constraints between image features and generated text. We believe that the attack formulation and evaluation framework introduced in this work provides a necessary foundation for developing and benchmarking such defenses in future studies.

---

> ### Author Response · Authors · 2026-01-29
>
> As the discussion period nears its end, we would like to follow up on our earlier reminder. We have submitted a revised manuscript and responses addressing all comments. Please let us know if any further clarification is needed.
>
> Thank you again for your time and attention.

---

> > ### Comment · Reviewer_uhyw · 2026-01-31
> >
> > The authors' rebuttal address most of my concerns. I will keep my original positive score.

---

### Official Review · Reviewer_4zxb · 2026-01-10

**Confidence:** 4
**Preliminary Rating:** 3
**Final Rating:** 4

**Summary:**

The paper introduces a new threat model to demonstrate how multimodal LLMs can be adversarially modified to fabricate clinical findings. The presented model is a targeted multimodal attack using joint perturbations of CT and text prompt embeddings. The results suggest high success rates for generating clinically dangerous CT reports, using the RadGEnome-Chest dataset.

**Strengths:**

1. The threat model is mostly clinically grounded. It is organ specific and generates anatomically plausible outcomes that are more reflective of clinical scenarios

2. The evaluation is comprehensive in some aspects and consists of complementary analyses of attack success rates, stealthiness, organ recognition and clinical finding detection

3. Safety is a critical aspect of clinical deployment and better understanding of these vulnerabilities is of value to the community

4. The paper is well written and clear

**Weaknesses:**

1. It is not clear to me if the proposed threat model is very practical. The assumption is white-box access to all model components, which somewhat undermines the motivation for adversarial perturbations, as they are typically used in black-box scenarios. An attacker with white box access can simply insert backdoors or alter the outputs directly without having to to bounded perturbations over 100 steps. This is not to say the analysis isn't interesting and valuable, but framing it as a somewhat practical threat model isn't obvious to me and could even be obfuscating the core findings themselves.

2. The evaluation is done on only one model and one dataset. It is not clear how this generalises to other models or datasets.

3. There could be better ablation of the attack components to see which ones are contributing the most to the results

4. No proper discussion or limitations sections. For a security-focused paper with real world implication, this is a pretty important omission. The assumptions for the practicality and threat model need to be acknowledged and discussed. Also important is the discussion of potential defences. Even if no defences are implemented, they need to be discussed and if no defence is possible, that needs to be stated as well, as that is a critical safety issue.

**Detailed Comments:**

- "Jean-Baptiste Alayrac eta l." should be "et al."
- The related work section discusses related work but could do a better job of placing the proposed method within the literature and discussing how it fits in

**Justification Of Final Rating:**

I thank the authors for addressing my core concern with the blackbox experiments and multi model evals.

However, one thing that should be clarified is whether text embeddings transfer. This is not obvious as they often live in a model specific embedding space, (if model to model transformations are not accounted for). If it's just the visual embeddings, that should be stated clearly.

Given these improvements, I have increased my score to 4.

**Justification Of The Preliminary Rating:**

The authors tackle an important problem in medical AI safety. The technical side is well explained and set up. The major issues stem from the framing (it reads like a realistic attack scenario, but is not very convincing) and the generalisability of the approach is also uncertain, given only one model on one dataset is tested.

**Questions To Address In The Rebuttal:**

1. Why would an adversary choose this attack method when they have white box access to the model? Is it meant to be realistic or a hypothetical case?

2. Are there any black box versions of this attack method? Or does it depend on white box access?

3. Have you tested on any other CT report gen models?

4. Is there a minimum epsilon below which the attack fails?

---

> ### Author Response · Authors · 2026-01-25
> **Official Comment by Authors**
>
> We thank the reviewer for their thoughtful and constructive feedback, and we are glad that they find the problem important to medical AI safety.
>
> **1. Practicality of threat model:**
>
> We thank the reviewer for the thoughtful feedback regarding the threat model. We would like to clarify that our white-box approach is intended to expose fundamental risks in 3D Medical MLLMs. Our intent in adopting a white-box setting is to isolate and characterize the vulnerability of the report-generation pipeline itself, independent of confounding factors such as query limits or partial access. We agree that unrestricted white-box access would trivialize attacks via direct output manipulation or backdoor insertion. However, our threat model considers a white-box but write-restricted adversary who has access to the model architecture and parameters (e.g., in a research or enterprise environment) and to patient CT data, but cannot modify model weights or post-hoc outputs due to integrity checks. In such realistic insider scenarios, the only feasible attack surface is the visual input and intermediate embeddings, motivating the study of bounded adversarial perturbations rather than direct output tampering. Furthermore, our framework has significant practical utility for Privacy-Preserving. By using our method, patients or data owners can apply **'adversarial cloaks'** to their CT images. These perturbations prevent unauthorized VLMs from extracting sensitive diagnostic information.
>
> To address concerns about black-box settings, we have conducted black-box transferability experiments. We evaluate black-box transferability by applying adversarial CT volumes optimized against a **Reg2RG model with a LLaMA-2-7B decoder (Chen et al., 2025b)** to a target model employing the same 3D-ViT encoder but a different language decoder **(Mistral-7B)**. During attack optimization, gradients flow through the entire pipeline, but the perturbation ultimately manipulates the visual and textual representations, not the visual encoder or language decoder parameters. We have added a section in our revised paper (**Section 5.3: Transferability of Adversarial CT Perturbations** and **Table 3**) that describes black-box transferability results.
>
> The attack achieves success rates of 59.4% and 38.7% when transferring adversarial CT images from LLaMA2-7B to Mistral-7B and vice versa, respectively. These findings underscore the practical risk of our threat model that an attacker possessing white-box knowledge of a commonly used medical vision encoder can effectively compromise a black-box system even if the specific language model remains proprietary or unknown.
>
> **2. Evaluation of proposed attack on other models and datasets:**
>
> We performed our evaluation on the CT-RATE dataset because it is currently the only publicly available large-scale repository of 3D chest CT volumes paired with narrative radiology reports. To address the concern about generalization beyond a single model, we extended our evaluation to **CT-CHAT** (Hamamci2024). Unlike our original baseline (Reg2RG), CT-Chat utilizes a fundamentally different architecture, including a LLaMA-3.1 8B language decoder and a CT-CLIP based visual encoder. We added the CT-CHAT model description in **Section 4.1** and the results are given in **Section 5.1** and **Table 1**. We have also evaluated the effect on NLG metrics when the CT-CHAT model is attacked to generate adversarial CT reports. The results for **NLG metrics** are shown in **Table 2**. The success of our attack on two state-of-the-art CT reporting MLLMs, which utilize different encoders and decoders, demonstrates that the vulnerability is model-agnostic and persists across modern 3D medical VLM designs.
>
> **3. Ablation study:**
>
> We would like to highlight that in our original paper, we already provided a component-level ablation of the proposed attack in Table 1. Specifically, we have analyzed the use of different attack surfaces (text embeddings in isolation and images in isolation) for Reg2RG model, revealing their respective contributions. In our **revised paper**, we have dedicated a separate section to this ablation study. Please refer to **Section 5.6 and Table 6** in our revised paper, where we analyze the use of individual (visual-only and text-only) and multimodal (visual and textual) perturbations on attack performance.
>
> **4. Effect of changing epsilon value:**
>
> We conduct an ablation study on the effect of varying the perturbation budget from 16/255 to 8/255 and 4/255, with the setup and results reported in **Appendix B (Table 7)**.
>
> **5. Limitations:**
>
> We have added a new section in our revised paper (**Section 6**) to discuss the limitations of our proposed attack.
>
> **6. Related work:**
>
> We revised the Related Work section to better position our method within prior work.
>
> **NOTE: All changes are highlighted in red text font in the revised paper.**

---

> ### Author Response · Authors · 2026-01-29
>
> As the discussion period nears its end, we would like to follow up on our earlier reminder. We have submitted a revised manuscript and responses addressing all comments. Please let us know if any further clarification is needed.
>
> Thank you again for your time and attention.

---

> ### Comment · Area_Chair_bGNM · 2026-01-31
> **Rebuttal discussion (optional)**
>
> Thanks again for your reviews. Discussion is a unique feature of MIDL that helps foster constructive engagement in the evaluation process, so if you have time, please feel free to look at the authors’ rebuttal, join the discussion, and update your final rating as you see fit. Thanks.

---

### Author Rebuttal · Authors · 2026-01-25

**Rebuttal:**

We thank all the reviewers for their insightful comments. We have replied to the comments and updated our paper accordingly to the best of our abilities. All changes made in the revised paper are **highlighted in red font**.

**Supporting Material:**

/attachment/d6d10b62348cd363858e86e1ad0cafbfc646982a.pdf

---

### Author Response · Authors · 2026-01-29
**Reminder from authors**

As the discussion period nears its end, we would like to follow up on our earlier reminder. We have yet to receive any engagement from reviewers **4zxb**, **uhyw** and **VJoT**. We have made every effort to address the concerns raised and remain open to further clarification. We would greatly appreciate your support in encouraging remaining reviewers to participate before the discussion window closes.

Thank you again for your time and attention.

Sincerely,

The Authors

---

### Meta-Review · Area_Chair_bGNM · 2026-02-02

**Recommendation:** Accept (Poster)
**Confidence:** 4

**Metareview:**

This paper studies clinically risky adversarial failures in chest CT report generation, focusing on scenarios in which high-risk findings are omitted or fabricated. Reviewers overall agreed that the problem itself is important and relevant, but differed in how heavily they weighed the paper’s limitations. Reviewer uhyw viewed the work positively, highlighting the clear and clinically grounded definition of harm and the convincing empirical evidence that current systems can be driven toward dangerous outputs, while also noting limitations such as reliance on automated extractors and the lack of human expert validation. Reviewers 4zxb and VJoT, on the other hand, expressed concerns about the realism of the white-box threat model, the limited evaluation in the original submission (a single model on a single dataset), and the incremental nature of the attack methodology, with VJoT placing particular emphasis on novelty. The AC agrees that these concerns are reasonable, and that the paper’s main contribution is not a new adversarial algorithm.

The AC finds that the authors responded substantively during the rebuttal and discussion period. They extended their evaluation to an additional CT report generation model, conducted black-box transferability experiments, added perceptual analyses and ablations, and introduced an explicit limitations section along with a discussion of potential defenses, addressing many of the core issues raised by the reviewers. While the threat model remains somewhat controlled and the methodological novelty is still limited, the AC believes the value of this work lies in its medical AI safety characterization and its clinically meaningful framing of harm, rather than in proposing a new adversarial technique. Overall, the paper offers insights that are worth discussing within the MIDL community. Balancing the differing reviewer perspectives, the AC believes the revised manuscript merits acceptance. The AC also commends the authors for their active engagement during the discussion period, which helped clarify the scope of the contribution and strengthened the final paper. In the camera ready version, the AC encourages the authors to clearly define what constitutes a clinically risky failure and to demonstrate why this is a concern in clinical practice with medical AI.

---

### Decision · Program_Chairs · 2026-02-14

Accept (Poster)